# Core Competencies of the Public Health Workforce in Climate Change and Extreme Weather Events Preparedness, Response, and Recovery: A Scoping Review

**DOI:** 10.3390/ijerph21091233

**Published:** 2024-09-18

**Authors:** Thierry Perreault-Carranza, Vivian Ni, Jonathan Savoie, Jacob Saucier, Joey Frenette, Jalila Jbilou

**Affiliations:** 1Faculté de Médecine et des Sciences de la Santé, Université de Sherbrooke, Sherbrooke, QC J1K 2R1, Canada; thierry.perreault-carranza@usherbrooke.ca (T.P.-C.); vivian.ni@usherbrooke.ca (V.N.); jonathan.savoie2@usherbrooke.ca (J.S.); jacob.saucier@usherbrooke.ca (J.S.); 2Centre de Formation Médicale du Nouveau-Brunswick, Moncton, NB E1A 7R1, Canada; joey.frenette@umontreal.ca; 3School of Psychology, Université de Moncton, Moncton, NB E1A 3E6, Canada

**Keywords:** public health, climate change, scoping review, extreme weather events

## Abstract

Climate change poses a significant threat to public health and safety, necessitating an urgent, coordinated response. Public health officials must be well-trained to effectively prepare for, respond to, and recover from extreme weather events. Despite emerging frameworks, a gap remains in their systematic application, risking future unpreparedness. This review aimed to identify the necessary competencies for public health professionals to manage climate change and the best methods to teach these skills. An academic librarian helped develop a keyword chain for a PubMed search, which included original articles and reviews concerning our research questions published in English or French between 1 January 2013 and 31 January 2024. Out of 255 potential articles, 31 were included in this scoping review. The results aligned with our objectives, revealing three main themes: core competencies, training and pedagogy strategies, and assessment approaches for public health professionals’ preparedness, responses, and recovery in the context of climate change and extreme weather events. This scoping review enabled us to provide a set of clear recommendations for future research and practice in training the public health workforce for managing extreme weather events and climate change.

## 1. Introduction

Climate change has been called the greatest threat to health in the 21st century [1]. It represents an ever-escalating global crisis with far-reaching implications, impacting not only the environment but also posing significant threats to public health and safety [1]. The correlation between climate change and the increasing frequency and severity of extreme meteorological events has been thoroughly documented [2,3,4,5]. As anthropogenic global warming ensues and we continue to obliterate temperature records, as evidenced in July 2023, the hottest month recorded in over 100,000 years, and with heat records broken on all continents through 2022 [1], one can expect that the impact of extreme weather events will be of unpreceded extent [6,7,8].

Individuals in Canada have lived through increasingly frequent extreme weather events as a result of global warming and climate changes. Among their provinces, the Maritime provinces of Canada, situated along the country’s eastern coastline, exemplify this vulnerability. Being rural, coastal regions, their geographical location casts a risk of increased vulnerability in the face of extreme weather events [9]. As global warming ensues, increasing atmospheric water vapor is bound to accentuate the severity of high precipitation, including floods, cyclones, and winter storms alike [10]. The provinces of Atlantic Canada are expected to undergo sea-level changes surpassing the global average [11]. Coupled with the already documented precipitation increases, the rising local sea levels are expected to elevate the frequency and intensity of extreme high-water-level events, resulting in heightened flooding and damage to both infrastructure and ecosystems [11]. The province of New Brunswick is already grappling with the socio-economic impact of climate change. A three-fold increase in disaster financial assistance programs highlights the growing burden [11]. In 2018 alone, floods incurred a cost of CAD 74 million for the government, a figure expected to rise in the coming years [11]. Moreover, these events can have a disastrous impact on populations, as hazardous exposure to coastal storms and floodings can lead to significant morbidity and mortality during their acute phases [12]. The consequences of these extreme weather events are, however, felt long after their departure. From displacement to residential destruction, storm-related stressors cast a significant burden on populational mental health [12]. Likewise, exposure to flood water and drinking water contamination are vectors for infection, while flood-damaged properties increase the risk of chronic allergies and respiratory conditions [12]. Atmospheric water vapor concentrations and high sea surface temperatures of anthropogenic nature are already known to have contributed to various extreme weather events, including the “Snowmageddon” of February 2010 [13], a record-breaking blizzard that paralyzed the northeast of America. In New Brunswick, Canada, the escalation in winter storm severity has already become a noteworthy phenomenon. In January 2017, an unprecedented ice storm struck the rural northeast of the province, leaving more than 133,000 homes and businesses without power or heating for up to two weeks. The extreme weather event wreaked havoc on infrastructure, and over 8000 citizens required immediate assistance [14]. Furthermore, ocean acidification, propelled by rising carbon dioxide concentrations in both the atmosphere and oceans, is poised to affect the marine ecosystems of the Maritime provinces [15]. This is particularly significant as their socioeconomic sphere relies heavily on fisheries [11]. While the Maritime provinces are particularly at risk from climate change and extreme weather events, the effects of these are felt across the country. For instance, in recent years, other provinces have seen an increase in various extreme weather events. For example, in 2023, Alberta saw wildfires that were about 10 times more intense than their normal average [16]. Moreover, British Columbia saw its deadliest natural disaster in province history in 2021, when a heatwave resulted in the deaths of 619 individuals [17]. While this extreme weather event was very difficult to respond and recover from, it is to be noted the province is predicting an increase in the frequency of similar heatwaves, according to recent climate simulations [17]. Thus, climate change is a significant issue that requires attention from all Canadian provinces and territories. 

While climate change has an impact in the Maritimes region and other regions of Canada, its effects are much more widespread. Indeed, climate change is an issue of global significance, which affects countries all across the world [1]. Humanity is grappling with the devastating repercussions of global warming on a worldwide scale; there is no room for delay in orchestrating a robust and all-encompassing public health response across the globe. *The Lancet*’s 2023 “Countdown on health and climate change” report clearly delineates the global impact of climate change and global warming [1]. Advanced detection and attribution science has revealed that over 60% of health-threatening high-temperature days in 2020 were more than twice as likely than not to have been due to human-induced climate change [1]. Heat-related deaths among individuals over 65 surged by 85%, surpassing the expected 38% increase. Meanwhile, climate change continues to harm natural and human systems critical for health [1]. The global land area affected by extreme drought rose from 18% in 1951–60 to 47% in 2013–22, affecting water security, sanitation, and food production [1]. Increased frequency of heatwaves and droughts in 2021 led to 127 million more people experiencing moderate or severe food insecurity compared with 1981–2010, posing risks of malnutrition and irreversible health effects [1]. As climate change progresses, extreme weather events have become increasingly frequent and common [1]. This rise in frequency poses significant risks to both the physical and mental health of populations worldwide [1]. In recent years, numerous extreme weather events, wildfires, droughts, and the spread of infectious diseases have been observed with greater frequency across the globe, resulting from climate change [1]. The impacts of these events have been felt worldwide, as evidenced by an analysis of social media trends, which revealed a peak in negative online publications related to extreme weather in 2022 [1]. Moreover, the economic losses from extreme weather events rose by 23% from 2010–14 to 2018–22, reaching USD 264 billion in 2022 alone, while heat exposure resulted in potential global income losses totaling USD 863 billion [1]. These findings underscore the urgent need for climate action to mitigate health risks, protect livelihoods, and address economic losses. To protect global populations from the increasing threats posed by climate change and extreme weather events, it is essential to implement effective measures that enable health systems and professionals worldwide to adapt and prepare [1].

From high-precipitation events including rising sea levels, storms, and floods, to low-precipitation events like heatwaves, droughts, and wildfires, to an exacerbation of disease through poor air condition and vector-borne infectious disease, the consequences of climate change are multifaceted and require a comprehensive and uniform response from public health authorities [1,18,19].

Amid the evolving landscape of climate-induced health risks, public health officials play a crucial role in preparing, responding to, and aiding recovery from climate change and extreme weather events [18,19]. The intricate nature of these challenges demands a transdisciplinary, coordinated effort among various stakeholders, including government agencies, non-governmental organizations, and local communities [11,19,20,21]. However, one of the significant challenges faced is the lack of cohesive coordination and alignment of interventions across these diverse entities [21]. This lack of synergy hampers the effectiveness of responses [21] and underscores the need for a comprehensive understanding of the core competencies required by public health officials and their collaborators to navigate these complex scenarios [22].

Competencies refer to the observable abilities of an individual to execute tasks or practices, which are manifested through actions or behaviors. These actions involve the integration of knowledge and skills at levels commensurate with the specific responsibilities assigned. In the context of medical education, particularly concerning climate change and extreme weather events, identifying and defining these competencies is of basic importance [22]. This is because effective public health workforce training should focus not only on theoretical knowledge but also on the practical application of skills and behaviors needed to navigate the complex challenges posed by climate-related events [23]. By assessing and ensuring the relevance and validity of these competencies across various training formats, education can better equip public health officials and professionals to effectively address the unique demands associated with climate change and extreme weather events.

Climate change affects all aspects of human activities globally, with its impacts on population health being among the most severe. Training public health professionals is crucial to ensure they can effectively perform essential public health functions such as surveillance, protection, prevention, and health promotion. In Canada, the Public Health Agency of Canada outlines a list of essential competencies for public health practitioners. However, these competencies do not currently address climate change, or the preparation, response, and recovery needed in relation to extreme weather events [24]. This lack of guidance and clear competencies for public health professionals poses a significant risk, as it may leave them unprepared to effectively respond to climate change and extreme weather events. Interesting frameworks have begun to emerge in the literature, attempting to delineate the principles or core competencies necessary for effective public health preparedness, responses to, and recovery from climate change events [22,25,26,27]. While there is partial agreement and alignment among these frameworks [25,26,27,28], a gap persists in the systematic application, assessment, and evaluation of these principles through continued medical education or other forms of training. This limitation has impeded the establishment of universally applicable and clearly defined core competencies and training goals. Therefore, it is crucial not only to identify the core competencies that enable the public health workforce to effectively contribute to health systems’ resilience (preparedness, response, and recovery) in the face of climate change and extreme weather events, but also to assess their applicability and validity across various forms of training. 

This article seeks to bridge these gaps through a scoping review that systematically examines the available literature to identify and synthesize the core competencies required for a public health workforce in the context of climate change and extreme weather events, including training strategies and curricula, as well as evaluation of learning outcomes. The results of this review are to be utilized to better inform the construction of a teaching program for healthcare professionals in New Brunswick, Canada. This program will aim to enhance preparation, responses, and recovery in the context of climate change and extreme weather events. However, we expect the results of this review and its recommendations to simultaneously become relevant for other provinces in Canada, as well as other countries globally.

## 2. Materials and Methods

Synthesizing the existing health literature is crucial for informed decision making and advancing clinical knowledge. Two main methods may be used: scoping reviews and systematic reviews. While both follow structured processes, they differ in their objectives and methodologies [29]. Scoping reviews generally have broader scopes and are conducted to identify knowledge gaps, clarify concepts, or investigate research conduct [29,30,31,32]. They also explore the extent, range, and nature of research on a particular subject, assess the necessity of conducting a systematic review, and compile existing research findings [32]. In contrast, systematic reviews are preferred when addressing clinically significant questions or providing evidence to guide practice. The choice depends on the research objectives and desired depth of analysis.

We elected to conduct a scoping review to explore the literature on core competencies essential for public health officials and collaborators in managing climate change and extreme weather events. Furthermore, we aimed to identify training practices and assessment tools, as well as to synthesize recommendations for future research and practice. This scoping review was conducted according to the Preferred Reporting Items for Systematic reviews and Meta-Analyses extension for scoping reviews (PRISMA-ScR) statement. The protocol for this scoping review has not been registered nor published. All relevant information describing the protocol is included in this manuscript. 

To achieve the study’s aim, the scoping review adhered to the JBI methodology for scoping reviews [33]. We proceeded according to the six-step process for conducting a scoping review recommended by Mak and Thomas [34]: (1) identifying the research question, (2) identifying relevant studies, (3) selecting studies to be included in the review, (4) charting the data, (5) collating, summarizing, and reporting the results, (6) optionally consulting stakeholders. The results of these steps are detailed in the subsequent sections.

### 2.1. Step 1: Identifying the Research Question

What are the core competencies essential for public health officials and their collaborators in the management of climate change and extreme weather events, and how can they be taught and evaluated for learning outcomes?

### 2.2. Step 2: Identifying Relevant Studies

The search was conducted using a specialized medical electronic database (PubMed). The keyword chain involved four key concepts: climate change, public health, training, and core competencies. The final search terms were the following: ((climate change) OR (extreme weather events) OR environment* OR (natural disaster) OR (global heating) OR (global warming) OR (climate crisis)) AND ((public health) OR (health professionals) OR doctor* OR physician* OR (medical specialist*) OR (healthcare worker*) OR nurse* OR (medical student*) OR (resident*)) AND (training OR education* OR webinar* OR cme OR (continued medical education) OR workshop*) AND ((core competencies) OR competenc* OR (professional skill*) OR skill* OR outcome*). A librarian specialist was consulted to validate the appropriateness of the keyword chain.

### 2.3. Steps 3 and 4: Selecting Studies to Be Included in the Review and Charting the Data

Articles were included if they met the following criteria: original articles, scoping reviews, or systematic reviews published in English or French between 1 January 2013 and 31 January 2024, describing core competencies or training programs and targeting the public health workforce, healthcare professionals, or medical and nursing students. Conversely, articles were excluded if they were conference abstracts, proceedings, or opinion articles, did not align with our research question, or did not meet any of the mirrored inclusion criteria outlined above. The selection step included two sub-steps: (1) review of the titles and abstracts and (2) review of the full texts (Table 1).

The initial search yielded 255 documents, all of which underwent screening. In total, 146 articles were excluded based on the review of titles and abstracts, and an additional 70 papers were discarded for various reasons: they did not align with our study’s objectives, they were opinion articles or duplicated studies, the did not target the desired audience, or they lacked published results. Finally, 38 articles were included in the review (Figure 1).

### 2.4. Step 5: Collating, Summarizing, and Reporting the Results

Each included document was individually reviewed to ensure its relevance to our research question. Furthermore, following the methodology of Mak and Thomas [31], a subset of articles underwent a double review procedure: ten percent of the papers were randomly selected and subjected to a two-step evaluation, during which they were independently reviewed by two researchers. We performed the double review process during the two selection sub-steps: title and abstract, and full text. In instances of disagreement regarding inclusion or exclusion, a third independent reviewer was consulted. Biweekly meetings were scheduled to discuss the process and plan the next steps. When needed, extra sessions were arranged to achieve a consensus. Relevant data, including the study’s objectives, descriptions of core competencies, geographic location of the study, theoretical framework and foundations, training strategies and programs, and evaluation tools, were extracted from each article and synthesized.

## 3. Results

### 3.1. Descriptive Statistics

The scoping review process resulted in a total of 31 articles selected and retrieved. As shown in Table 2, 6 papers were literature reviews (including 2 systematic reviews, 1 systematic narrative review, 2 narrative reviews, and 1 scoping review), while the remaining 25 articles were single studies. 

In recent years, there has been an evident increase in research focusing on climate change and health education, as evidenced by the distribution of publication dates in Figure 2. Nearly half of the articles included in our study were published in 2023. The studies included in the review were conducted across diverse geographical locations, spanning five continents. Most of the studies were conducted in the United States of America (41%), while 16% involved multiple countries.

The target populations of the reviewed studies involved various groups, with healthcare professionals being the most common, appearing ten times. Medical students were referenced eight times, followed by physicians with five mentions. Nursing students and public health students were less frequently encountered, with four and three instances, respectively. Finally, university students were mentioned only once. No study specifically targeted the public health workforce. 

This scoping review identified various climate-related themes. Many themes overlapped across articles and were presented as climate change modules or sub-competency learning objectives. These themes were grouped into 13 major categories, as detailed and listed in Table 3, ordered by frequency. The most frequently encountered themes included the fundamental science of climate change and health consequences of climate change.

### 3.2. Data Synthesis and Main Findings

Our research findings are summarized and presented in accordance with our research objectives: (1) core competencies; (2) training programs and pedagogy strategies; and (3) assessment approaches to inform future competency-based education programs for public health workforces in climate change and extreme weather events preparedness, response, and recovery.

### 3.3. Core Competencies 

Among the 31 articles selected for our study, 21 discussed core competencies considered essential to integrate into curricula or training programs, enabling students/trainees to address climate change in their future practice. Of these, 15 articles relied on pre-existing frameworks, while 6 articles developed a curricular competency model tailored to their study objectives.

Nine studies implemented competency frameworks in education courses on climate change and extreme weather events, while six papers recommended these frameworks for such courses but did not use them in their own research (Table 4). These included the Global Consortium on Climate and Health Education (GCCHE) core competencies [49,55,59], the American Association of Colleges of Nursing (AACN) essential domains [60,61], the Planetary Health Education Framework core domains [60], and the public health emergency preparedness domains [44].

On the other hand, six studies developed competency frameworks to address the challenges of climate change and its impact on public health (Table 5). Jagals and Ebi (2021) [22] proposed a six-domain framework for healthcare workers encompassing environmental science, upstream drivers, evidence assessment, risk management, mitigation strategies, and collaboration. Another four-domain competency framework was developed by Lemery et al. (2019) [62], covering fluency in climate-health impacts, proficiency in mitigation and adaptation concepts, leadership in climate and health program development, and effective science communication. In addition, Philipsborn et al. (2021) [56] highlighted a climate and health curricular framework aligned with ACGME core competencies, focusing on climate change knowledge, clinical adaptations, and healthcare delivery implications. Furthermore, Arora et al. (2023) [39] divided competencies into four tiers ranging from awareness and knowledge building to tailored skill development, whereas Valois et al. (2016) [51] highlighted knowledge domains covering climate change, as well as heat-related, vector-borne, and water-borne diseases. Lastly, the One Health model developed by Rabinowitz et al. (2017) [63] uniquely integrates a triad of animal, environmental, and human health systems. It emphasizes interdisciplinary communication, zoonotic disease recognition, and ecosystem health understanding.

### 3.4. Training and Pedagogy Strategies

This scoping review comprehensively assessed the training programs and pedagogy strategies mobilized to better equip students/trainees with climate change core competencies. Among the 31 articles retrieved, 20 explored training methods and strategies. We synthesized findings from 15 single studies, two systematic reviews, two scoping reviews, and one narrative review (Table 6 and Table 7).

Among the fifteen original articles examined, online learning emerged as the prominent delivery method, with ten articles opting for this mode. More specifically, six opted for a synchronous approach and broadly described a variety of teaching methods used, including didactic lectures or online modules, and more interactive approaches such as question-and-answer sessions, group work, and simulations [40,42,43,47,49,58]. Two articles opted for an asynchronous approach and described online learning with self-paced modules [44,46]. A single article described online learning in both approaches [45]. Additionally, one article initially selected in-person training but had to transition to an online synchronous approach in response to COVID-19 restrictions [41].

In contrast, in-person teaching was observed in five articles, incorporating a wide variety of methods such as didactic lectures, practice scenarios, and workshops. Notably, two studies [57,64] implemented a train-the-trainer approach, aiming for trained participants to subsequently facilitate a similar session themselves and adopt curriculum modifications in their own institutions. One article adapted its training approach to the four-phase model of interest development and the action competence framework. The former promotes situational interest by linking new information to personal and professional connections, while the latter fosters the development of action competences through discussions on climate change including identifiable impacts and specific needs relating to the participants’ region [44].

Additionally, among the fifteen articles, three of them discussed courses integrated into a health science curriculum [45,54,55]. Furthermore, Shea et al. (2020) [48] conducted a cross-sectional study of health educational institutions and provided insights into various teaching methods practiced, including laboratories, didactic lectures, in-class exercises, and synchronous online learning methods through tutorials and massive open online courses.

Review articles provided further depth into educational practices within climate change education. Similarly to our small review, Evans et al., (2023) [59] found a significant variability in the delivery methods of climate change-related continuing education opportunities in the United States. Most programs identified were also offered virtually, although, contrary to ours, they were mainly asynchronous classes. Boekels et al. (2023) [38] discussed teaching methods used in the field of climate teaching in medical studies, highlighting a large spectrum of approaches, from lectures to problem-based learning. They also described an international effort focused on the integration of this topic into medical curricula, by embedding in existing teaching formats or teaching climate change separately. Bray et al. (2023) [35] reported educational approaches used in sustainable healthcare education, including workshops, clinical skills sessions, and simulation-based training. Lopez-Medina et al. (2019) [36] identified distinct changes in nursing curriculum reform in higher education, enabling interdisciplinarity. The movement was from didactic learning to more progressive approaches such as reflective activities and a learning environment that moved beyond the classroom. Lastly, Arora et al. (2023) [39] reviewed climate and health courses offered in accredited schools of public health in the United States. Some courses primarily focused on climate and health while others integrated this topic into a broader course and included both online and in-person forms of delivery.

### 3.5. Assessment Approaches

Out of the 31 articles selected for our study, 20 discussed assessment tools and post-training evaluations. Seven studies applied assessment methods including graded assignments, quizzes, poster presentations, peer review group assignments, online discussion board assignments, attendance, essays, posed research questions, multiple choice tests, end-of-modules tests, and writing an op-ed as a final project (Table 8). Five studies used self-reported assessments approaches (Table 9).

Six studies did not apply any assessment tools, but instead provided recommendations for evaluation methods (Table 10). Some of these recommendations include identifying clear intended outcomes from the onset of delivery, using both quantitative and qualitative evaluation methods, and assessing higher cognitive skills as well as factual knowledge. They propose methods such as multiple-choice questions, oral questions, objective structured clinical examination (OSCE), clinical evaluations, chart audits, direct observation, standardized patient checklists, small-group discussion, final presentations, reflective writing, special diaries, essays, and complex project work.

Six studies evaluated trainee satisfaction and surveyed participants to evaluate whether the training reached the specified learning objectives. The details are listed in Table 11.

## 4. Discussion

### 4.1. Target Population

Taking into consideration that our primary objective was the identification of core competencies, training programs and pedagogical strategies and assessment methods required for the public health workforce to prepare for, respond to, and recover from the consequences of climate change and extreme weather events, the authors were particularly interested in the identification and analysis of research with the public health workforce as the principal study population. Nonetheless, this scoping review highlights the paucity of research regarding the training objectives, core competencies, and past training events for the public health workforce. 

Numerous core competencies and frameworks have been proposed for healthcare professionals in the context of planetary health and climate change. This review identified six studies that designed competency models with the objective of responding to climate change [20,48,53,57,59,60]. The target audience of these models is large, encompassing medical and nursing students, nurses, physicians, and the globality of health professionals. It is important to highlight that none of the articles reviewed specifically addressed core competencies designed for the public health workforce in meeting the challenges posed by climate change. Considering this gap of knowledge in the literature, there is an imperative need to establish and verify a universally accepted set of core competencies designed specifically for the public health workforce. These competencies would play a pivotal role in equipping professionals to effectively prepare for, respond to, and recover from the impacts of climate change and extreme weather events. Numerous core competencies and frameworks have been proposed for healthcare professionals in the context of planetary health and climate change. 

Similarly, this review highlighted a multitude of training events and continued medical education sessions (*n* = 15) aimed at enhancing the knowledge of nurses, medical students, and health professionals in addressing the challenges posed by climate change and preparing them to respond effectively to its consequences. Surprisingly, however, this scoping review did not uncover any continued medical education events specifically designed for the public health workforce. Recognizing the pivotal role played by the public health workforce in the preparation for, response to, and recovery from climate change and extreme weather events, it becomes crucial to develop tailored training modalities to address their unique needs. To address the identified gap in the literature, future research should focus on developing and evaluating a continued medical education program. This program should be based on a framework of core competencies specifically tailored to the needs of the public health workforce and enhance the preparedness, response, and recovery capabilities of the public health workforce in the context of climate change and extreme weather events.

### 4.2. Core Competencies and Competency Frameworks

The integration of climate change-related competencies into healthcare education is essential to prepare future professionals to address the challenges posed by environmental changes. Our review highlights several existing frameworks, such as the GCCHE Core Competencies, the National Competence-Oriented Catalogue of Learning Objectives in Medicine, and the AACN Essentials: Core Competencies for Professional Nursing Education (Table 4). Common competencies among these models include effective communication, basic clinical skills, recognizing the interconnectedness of nature, and promoting equity and professional responsibility. 

However, our analysis reveals a need for frameworks to facilitate the intersection of climate change and healthcare. While several of the articles provide examples of how to acquire the competencies, they do not detail how to integrate these methods into healthcare curricula. Models like “Climate Change and Medical Education: An Integrative Model” and “Development of a Course on Complex Humanitarian Emergencies: Preparation for the Impact of Climate Change” offer key insights for guiding medical schools to integrate climate change education into medical training.

Six studies designed competency models aimed at tackling climate change and its effects on public health [22,51,56,60,62,63]. These frameworks highlight the importance of climate change knowledge, mitigation, and adaptation strategies, as well as effective communication. Among them, five studies focused on human factors [22,39,51,56,62], while one study’s model [63] revolved around animal health considerations, and all presented well-developed frameworks with a minimum of three domains, some with up to six (Table 5). Each paper outlines its model in detail and breaks down each domain into specific learning objectives or sub-competencies. Four out of the six texts provide methods for achieving these competencies, such as seminars, courses, lectures, clinical rotations, role-playing, standardized patients, and shadowing veterinarians [39,56,63]. Moreover, Lemery et al., 2019 [62] suggest that participation in climate health projects with various organizations, teaching opportunities, podium presentations, research projects, and extensive clinical experience can also help attain these competencies, particularly for climate and health science fellows. They also explain the underlying principles guiding their framework development, drawing from sources such as a literature review [22], resources from the emergency medicine department at the University of Colorado [62], the ACGME core competencies [56], and the core competencies outlined by the Association of Schools and Programs of Public Health (ASPPH) and GCCHE [39]. 

Moving forward, future research could explore the implementation of these competency frameworks into healthcare curricula, as well as their effectiveness in terms of student learning outcomes. Additionally, there is need to develop an integrative theoretical framework with the aim of unifying these reference frameworks to create a global and comprehensive reference framework. 

### 4.3. Pedagogy

The findings of this scoping review reveal a substantial variety in teaching methods, as outlined in Table 6 and Table 7. Notably, no singular approach emerges as a clear favorite. Moreover, the target populations for these methods include a wide spectrum of health professionals, with a noticeable lack of specific focus on the public health workforce. Evans et al., 2023 [22,39,51,56,62,63] underscore this issue, noting in their review that out of twenty identified continuing medical education programs, only three were specifically tailored for the public health workforce. Similarly, among the fifteen articles we examined, only one was directed towards this population. Significantly, the absence of randomized control trials and the high prevalence of quasi-experimental designs present challenges for researchers aiming to identify the most effective teaching method. Furthermore, the lack of standardization in training tools further complicates the evaluation process. 

It is noteworthy that the most prevalent teaching delivery method observed was online learning, mainly synchronous through lectures and online modules This prevalence may be attributed, at least in part, to the constraints imposed by COVID-19 restrictions, as evidenced by the temporal distribution of these publications that were mainly published between 2020 and 2024. In-person training was less frequent, remaining predominantly common within teaching from formal health curricula. However, there is currently no conclusive evidence regarding the comparative effectiveness of these delivery methods. 

The current landscape of educational programs falls short in meeting the demands of healthcare professionals who require education on climate change to prepare them to respond effectively to its consequences. Among 2817 medical schools across 108 countries surveyed, just under 15% offered courses specifically addressing climate change, according to Boekels et al.’s (2023) narrative review [38]. Similarly, Arora et al. (2023) published a scoping review of public health curricula in the United States and noted that a significant portion (51%, or 46 out of 90 programs) did not include any climate change-related courses [39], illustrating a clear gap in teaching on this crucial topic. Many articles described frameworks for integrating planetary health (i.e., an action-oriented multidisciplinary social movement focusing on understanding and addressing the effects of human-induced changes to the Earth’s natural systems on human well-being) into existing health curricula [55,56,60,61,62,63]. These frameworks often revolve around competency-based approaches and could serve as valuable guides for implementation. One commonly cited challenge is the perceived burden on already overloaded medical programs, making it difficult to incorporate additional training on climate change and planetary health. However, integrating this content into existing courses rather than creating entirely new ones could offer a more feasible solution, ensuring that vital topics are addressed within the current educational curriculum. 

In this context, virtual reality (VR) presents a compelling avenue for revolutionizing planetary health education by offering immersive, experiential learning opportunities. Through VR simulations, learners can engage with dynamic environments, experiencing firsthand the interconnectedness of climate change and human health through simulated scenarios. Future research avenues could conduct cross-sectional surveys or assessments among public health workers to gauge their needs, expectations, and preferences. This approach holds promise in providing valuable insights into the specific requirements of this critical workforce and can inform the development of tailored training programs. 

### 4.4. Evaluation Methods

Among the studies mentioning assessments (Table 8, Table 9, Table 10 and Table 11), the majority either assessed the satisfaction of the participants or relied on self-reported learning. Given that training on climate change and extreme weather events for the public health workforce is a relatively new concept, the lack of standardized evaluation formats may stem from the emphasis on learning objectives, core competencies, and pedagogy formats within the field.

While a variety of assessment methods have been utilized in studies on climate change and extreme weather events, there remains a scarcity of research exploring which methods are most effective for assessing such a broad and complex topic. Boekels et al., 2023 [38] recommend evaluating deeper knowledge through non-traditional formats like presentations, reflective writing, essays, or complex projects. In alignment with this perspective, we propose further exploration into acquiring core competencies through dynamic assessment methods such as OSCE, direct observation, simulations, and oral questions. However, our scoping review did not identify any specific assessment tools mentioned within the studies. Future research should focus on developing diverse evaluation instruments tailored to evaluate the various core competencies required in this field.

### 4.5. Strengths and Limitations

The present scoping review presents several strengths. Firstly, to our knowledge, it is the first study to focus specifically on climate change and the public health workforce. It provides a structured overview of the existing literature, helping people understand the breadth and depth of available evidence on the topic. Secondly, the selection of our studies and the extraction of data were systematic, allowing us to control for bias selection. The keyword chain was validated by an expert librarian and the selection of articles followed the PRISMA 2020 recommendations. Additionally, the inclusion of articles from various geographical locations, spanning five continents, further enhances the review’s thoroughness.

However, our study was conducted solely on the PubMed database, suggesting the need to explore other databases, such as Embase, for a more comprehensive literature review. Another limitation is that the present review only considers articles published in the last 10 years. Extending the search period to the last 20 years may be pertinent given the increasing interest in climate change and the public health workforce. Finally, given that climate change and extreme weather events are global phenomena, our inclusion criteria of only French or English articles may have excluded some potentially relevant studies.

## 5. Conclusions

In conclusion, our study primarily sought to identify the core competencies needed for healthcare professionals to effectively manage climate change and extreme weather events. Additionally, we aimed to identify training practices and strategies and assessment and evaluation tools as well as derive recommendations for future research and practice in terms of extreme weather events management training for the public health workforce.

## Figures and Tables

**Figure 1 ijerph-21-01233-f001:**
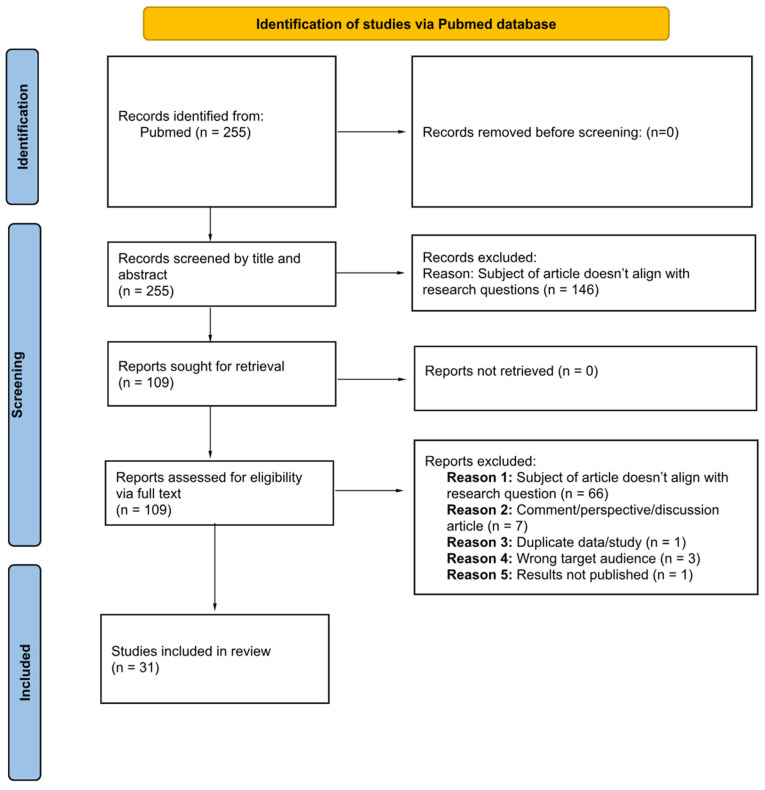
PRISMA flow chart.

**Figure 2 ijerph-21-01233-f002:**
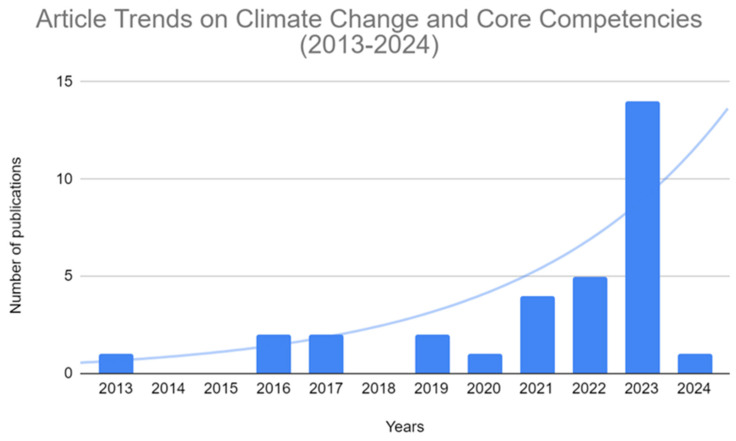
Number of articles on core competencies published between 1 January 2013 and 31 January 2024.

**Table 1 ijerph-21-01233-t001:** Summarizing the selection criteria.

Inclusion Criteria	Exclusion Criteria
Original articles, scoping reviews, or systematic reviews	Conference abstracts, proceedings, perspectives, opinions, and commentary articles
Published in English or French language	Subject of article does not align with our research question
Published between 1 January 2013 and 31 January 2024	
Describes core competencies or training program	
Targets healthcare professionals or medical and nursing students	

**Table 2 ijerph-21-01233-t002:** Literature reviews included, study counts, and period covered (oldest article year and most recent article included in the review).

Literature Reviews	Study Counts	Period Covered
Sustainable healthcare education: A systematic review of the evidence and barriers to inclusion [35]	23	2008–2021
Competencies on environmental health and pedagogical approaches in the nursing curriculum: A systematic review of the literature [36]	32	2004–2017
Are we teaching the health impacts of climate change in a clinically relevant way? A systematic narrative review of biomechanism-focused climate change learning outcomes in medical curricula [37]	22	2014–2023
Climate change and health in international medical education—a narrative review [38]	50	2014–2021
Core competencies for health workers to deal with climate and environmental change [22]	2 ^a^	2015–2018
Assessing climate and health curriculum in graduate public health education in the United States [39]	38 ^b^	2018

^a^ This study included a narrative review of 212 articles and a scoping review of 53 articles. ^b^ This study assessed available online public health graduate course syllabi.

**Table 3 ijerph-21-01233-t003:** Major themes in order of frequency.

Theme	Description
1. The fundamental science of climate change [22,37,40,41,42,43,44,45,46,47,48,49,50,51,52,53]	Contribution of health systems to climate change. Changing current and future geographic ranges and seasonality. Weather, climate variability, and environmental relationships. Natural and anthropogenic changes. Methods or tools to analyze health risks and climate information.
2. Health consequences of climate change [22,37,38,39,40,41,42,43,44,45,46,47,48,49,50,51,54,55,56,57]	Environment–health interaction; acute and chronic stressors. Impacts of climate change on health systems and delivery.
3. Transformative actions in climate change [28,37,45,46,47,48,49,50,52,53,54,56,57,58]	Sociopolitical level. Individual level: climate-friendly behaviour. Infrastructural level: public transport, bike lanes. Adapting clinical practice for climate change: values, leadership, and advocacy around climate change awareness and mitigation. Public narrative, intentional relationships, developing leadership structures, strategies to identify resources, and securing commitments. Sustainable health care system. Community involvement in climate and health initiatives.
4. Extreme heat and vulnerable populations [40,42,43,45,47,50,51,54,55,56]	Global warming and wildfires. Heat stress. Vulnerability of patients.
5. Climate change communication [42,45,46,48,49,52,53,54,57,58]	Climate-sensitive health counseling in practice. Written and oral communication skills with stakeholders in government and other sectors. Inter and transdisciplinary collaboration.
6. Emergency preparedness [28,40,43,44,45,47,48,52,53]	Management, mitigation, and adaptation. Resilience and recovery. Infrastructure/disaster preparedness response and delivery systems.
7. The impacts on mental health [40,42,43,45,47,51,54,56]	Mental health illnesses. Consequences of trauma/natural disasters.
8. Extreme weather events [40,42,45,47,51,55]	Extreme precipitations, floods, hurricanes, and tropical storms. Drought and water scarcity. Heatwaves.
9. Water and vector-borne disease [42,43,45,47,51,55,56]	Infectious disease sensitive to climate change. Zoonotic diseases.
10. Air quality and health [40,42,43,45,54,56]	Respiratory illness and allergies.
11. Health equity, justice, and ethics [28,48,50,52,58]	Intersections of global warming, urbanization, poverty, and access to care. Social and environmental justice issues. Health disparities.
12. Food security [42,43,45,56]	Malnutrition and food insecurity.
13. Hazard and exposure pathways [22,56]	Injuries and toxic exposures.

**Table 4 ijerph-21-01233-t004:** Recommended and implemented competency frameworks in health care for climate change and extreme weather events.

Article	Competencies
Fülbert et al., 2023 [54]	National Competence-Oriented Catalogue of Learning Objectives in Medicine (Germany, practical skills, and medical competencies for medical undergraduate curriculum (the competence level includes factual knowledge, know-how, action competence)).
Rogers et al., 2023 [44]	The Action Competence Framework by Jensen and Schnack (1997) (Denmark, conceptual framework describing the concept of action competence); The public health emergency preparedness domains (2007) (USA, encompassing six key preparedness domains: community resilience, incident management, information management, countermeasures and mitigation, surge management, and biosurveillance).
Flaten et al., 2023 [60]	Planetary Health Education Framework core domains (Canada, for nursing education programs, encompassing interconnection within nature; anthropocene and health; systems thinking and complexity; equity and justice; movement building and systems change); The AACN Essentials: Core Competencies for Professional Nursing Education (USA, for nursing education programs, including knowledge for nursing practice; person-centered care; population health; scholarships for nursing; quality and safety; interprofessional partnerships; systems-based practice; informatics and healthcare technologies; professionalism; personal, professional, and leadership development).
Navarrete-Welton et al., 2022 [55]	GCCHE Core Competencies (USA, for climate and health education, encompassing knowledge and analytic skills; communication and collaboration; policy; public health practice; clinical practice)
Griffin et al., 2022 [61]	The AACN Essentials: Core Competencies for Professional Nursing Education
Lopez-Medina et al., 2019 [36]	No competency framework is discussed, but rather general competencies are proposed.
Boekels et al., 2023 [38]	CSH Networks (UK, key competencies in sustainability, including systems thinking competence; futures thinking (anticipatory) competence; values thinking (normative) competence; strategic thinking competence; interpersonal (collaboration) competence; integrated problem-solving competence.)
Evans et al., 2023 [59]	GCCHE core competencies
Basu et al., 2024 [58]	The structural competency framework by Neff et al., 2020
Cadet 2022 [45]	NONPF (USA, core competencies for nurse practitioners, not specific to climate change or EWEs)
Rublee et al., 2021 [47]	AFEM
Sorensen et al., 2023 [49]	GCCHE core competencies
Simon et al., 2023 [28]	“Cross-cutting principles for PHE” by Stone et al. (2018) “A Framework to guide PHE” by Guzmán et al. (2021) “The AMEE consensus statement: Planetary health and education for sustainable healthcare” by Shaw et al. (2021)
Teherani et al., 2023 [57]	Mention of “Faculty development and partnership with students to integrate sustainable healthcare into health professions education” by Tun et al., 2020. No competency framework discussed in detail.
Wabnitz et al., 2022 [52]	National Competence-Oriented Catalogue of Learning Objectives in Medicine

AACN: American Association of Colleges of Nursing; AFEM: African Federation for Emergency Medicine; AMEE: Association of Medical Education in Europe; CSH: Centre of Sustainable Healthcare; GCCHE: Global Consortium on Climate and Health Education; NONPF: National Organization of Nurse Practitioner Faculties; PHE: Planetary Health Education.

**Table 5 ijerph-21-01233-t005:** Self-developed competency frameworks with descriptions.

Articles	Core Competencies Frameworks Developed De Novo
Jagals and Ebi, 2021 [22]	Six domains: 1. Climate, environmental change, and associated health sciences; 2. Upstream drivers of climate and other environmental changes; 3. Evidence, projections, and assessments; 4. Iterative risk management; 5. Mitigation, adaptation, and health co-benefits; 6. Collective strategies—harnessing international/regional/local agreements and frameworks.
Lemery et al., 2019 [62]	Four domains: 1. Fluency with climate and health impacts: understanding how these perturbations in earth science impact human well-being—both pathophysiologic and societal; 2. Facility with concepts of mitigation and adaptation as actions within public and private entities, and evaluation of the quality and effectiveness of such actions related to health impacts; 3. Capacity to lead effective climate and health programmatic development within the academic, public, and private sectors; 4. Outstanding science communication skills to effectively articulate the impacts of climate change upon human health—both in academia and through lay communication.
Philipsborn et al., 2021 [56]	Three domains: 1. Knowledge of climate change and its effects on health; 2. Climate change-related adaptations for clinical practice; 3. Implications of climate change for healthcare delivery.
Rabinowitz et al., 2017 [63]	One Health competencies model: Skill sets: • Ability to elicit a history of human–animal–environment interactions; • Inter-professional communication and teamwork skills; • Ability to recognize and treat zoonotic and vector-borne disease; • Ability to assess and improve patient environments. Knowledge competencies: • Zoonotic and vector-borne diseases; • Animals as sentinels; • Human–animal bond and role of service animals, therapy animals, etc.; • Prevention of animal-related injuries; • Ecosystem function and health; • Food systems, in particular animal-source foods, in human health and disease; • Role of environment on human health; • Ethics and values including balance of health and environmental values and legal/ethical limits for physicians dealing with veterinary issues and veterinarians dealing with human health issues; • Comparative clinical and evolutionary medicine.
Arora et al., 2023 [39]	Four tiers: 1. Building awareness; 2. Building knowledge; 3. Enhancing knowledge, problem- olving, and critical thinking; 4. Tailored skill building.
Valois et al., 2016 [51]	General knowledge about the following: 1. Climate change; 2. Heat-related illnesses; 3. Climate change, extreme weather events, and modification of vector-borne and zoonotic diseases; 4. Climate change, extreme weather events, and modification of water-borne diseases; 5. Mental health impacts of natural disasters.

**Table 6 ijerph-21-01233-t006:** Training programs and pedagogy strategies from single studies.

Single Studies	Target Audiences	Study Design	Format	Methods of Teaching	Additional Information
Dunne et al., 2022 [46]	Medical students	Quasi-experimental before-and-after study	Asynchronous online learning	Four self-paced online modules, each followed by a quiz with detailed explanations of the answers including links to further reading.	
Sorensen et al., 2023 [49]	Health professionals	Quasi-experimental before-and-after longitudinal survey study	Synchronous online learning	Ten didactic lectures followed by a question-and-answer session. Also included bi-monthly interactive skills and practice sessions structured around clinical cases, climate tools, communication, and leadership strategies.	Access to syllabus, resource bank, and reading
Basu et al., 2024 [58]	Health professionals	Quasi-experimental before-and-after survey study	Synchronous online learning	Six 2 h monthly sessions and 3 weekend immersions that lasted 4 h on both Sat and Sun. Didactic and discussion sessions, pre-work videos and readings, small group workshops, extensive coaching, and self-reflection exercises.	Longitudinal fellowship program
Cadet, 2022 [45]	Family nurse practitioner students	Non-experimental design	Synchronous and asynchronous online learning	14-week course including lectures, readings, discussions, case studies, simulations, team-based learning, projects, and quizzes. Nine interactive online modules.	Part of a nursing curriculum
Charlesworth et al., 2013 [64]	Trainees in public health medicine and other medical practitioners	Quasi-experimental before-and-after survey study	In-person workshop.	A 4–5 h session given by invited guest speakers. Train-the-trainer approach.	
Teherani et al., 2023 [57]	University faculties of health sciences	Quasi-experimental before-and-after survey study	In-person workshop.	A single daylong training session, including small-group activities. Train-the-trainer approach.	Online repository of educational resources available
Katzman et al., 2023 [40]	Healthcare professionals	Quasi-experimental before-and-after survey study	Synchronous online learning.	22-week course, weekly 60 min online sessions. Evidence-based lectures, question-and-answer session, live simulated cases, skills session on available climate mapping and data access tools for educational use with patients, evidence-based references provided in the chat during sessions.	Access to the presentation and course material
Jonas et al., 2023 [41]	Medical students	Quasi-experimental before-and-after survey study	In-person or synchronous online learning.	A single 45 min online or in-person interactive seminar. An initial lecture followed by an interactive role-playing session and small-group plenary session, concluded by a lecture.	
Fülbert et al., 2023 [54]	Medical students	Quasi-experimental before-and-after survey study	In-person course	Nine 90 min seminars given once a week, with one concluding six-hour session. Didactic approach characterized by active participation through discussion, teaching at eye level, combining theoretical knowledge, clinical case studies, and practical units on successful interviewing.	Optional elective course
Katzman et al., 2022 [42]	Healthcare professionals	Quasi-experimental before-and-after longitudinal survey study	Synchronous online learning	Eight-week course, weekly 75 min online lectures with various teaching methods such as evidence-based lectures, question and-answer session, live simulated cases, skills session on available climate mapping and data access tools for educational use with patients, evidence-based references provided in the chat during sessions.	Free access to the presentation and course material
Floss et al., 2021 [43]	Healthcare professionals and opened to a wider non-healthcare audience	Quasi-experimental before-and-after study	Synchronous online learning	Online modules were made available weekly. Modules were supported with book suggestions, relevant articles, videos, case discussion, peer project review and interactive infographic.	
Rogers et al., 2023 [44]	Healthcare professionals	Design-based research	Asynchronous online learning	8-week course. Pedagogy approach based on the four-phase model of interest development (2006) and the action competence framework (1997).	The course content is available in the Appendix A of the article
Navarrete-Welton et al., 2022 [55]	Medical students	Descriptive study	In-person	One single 1 h training session, including a didactic lecture with practice scenarios. Optional planetary health elective course over 8 didactic lectures.	Part of a nursing curriculum
Shea et al., 2020 [48]	Health professions students	Descriptive cross-sectional study	In-person and synchronous online learning	Education institution offered in-person training as part of non-required and required core courses through various teaching methods such as laboratories (*n* = 1), didactic lectures (*n* = 24), in-class exercises (*n* = 20), and internships outside the classroom (*n* = 2). Additionally, synchronous online learning methods through tutorials and massive open online courses (*n* = 8).	
Rublee et al., 2021 [47]	Medical students, residents/registrars, medical officers, nurses, pre-hospital providers, educators, and researchers	Non-experimental design	Synchronous online learning	A single full-day session that included didactic lectures, focused discussions, and a forum to discuss ideas.	

**Table 7 ijerph-21-01233-t007:** Training programs and pedagogy from review studies.

Review Study	Target Audience	Study Design	Teaching Formats
Evans et al., 2023 [59]	Healthcare professionals	Scoping review	Continuing medical education programs. Nearly exclusively in the online format (*n* = 20). Variability among synchronous (*n* = 2), asynchronous (*n* = 15), and hybrid (*n* = 3) formats. Most programs were self-paced structured programs that offered certificates or certificates of completion.
Boekels et al., 2023 [38]	Medical students and physicians	Narrative review	Noted a variety of delivery methods of climate change-related courses, including didactic lectures, project work, and reflective journals.
Bray et al., 2023 [35]	Healthcare professionals	Systematic review	Health professions curricula. Workshops (*n* = 7), scenario-based clinical skills sessions (*n* = 6), modules (*n* = 3), seminars (*n* = 2), simulation-based training (*n* = 1) and a photograph-based reflective exercise following a field trip (*n* = 1). Specific pedagogical methods employed within workshops included presentation, large-group discussion, small-group activities, guest speakers, case-based learning, role-play exercises, and videos. Six studies included preparation, mostly pre-reading (*n* = 5), although virtual cases, multimedia resources, online forums, and podcasts were utilised (*n* = 1 for each).
Arora et al., 2023 [39]	School of Public Health students	Scoping review	Climate change-related course included graduate-only and joint undergraduate/graduate courses, and both online and in-person forms of delivery. Courses either focused on or integrated climate change and health.
Lopez-Medina et al., 2019 [36]	Nurses	Systematic review	Nursing curriculum. A range of educational approaches were discussed. Developing interdisciplinary and flexible education methods; diversifying the space in which students are taught; student-centered learning and multidisciplinarity; diversifying educational activities by combining traditional methods such as lectures with others like simulations or discussion sessions. Identified the need to enhance delivery of information through cross-disciplinary work.

**Table 8 ijerph-21-01233-t008:** Studies using assessment methods post-climate-related events training.

Article	Assessment Tools and Evaluation Methods
Cadet, 2022 [45]	Students were evaluated though various means including graded assignments, quizzes, poster presentations, peer review group assignments, and online discussions.
Sorensen et al., 2023 [49]	Assessments involved a survey at the end of the training and attendance tracking, along with a post-training questionnaire to gauge course effectiveness.
Fülbert et al., 2023 [54]	Student performance was assessed through an essay on an individually posed research question or a seminar summary focusing on pertinent aspects of patient consultations and two recent publications on the topic.
Floss et al., 2021 [43]	Participants underwent assessment via multiple-choice question tests after each module and an environmental action-plan essay at the end of the course.
Rogers et al., 2023 [44]	Students were asked to produce an ungraded end-of-course assessment. However, detailed information was not provided.
Lemery et al., 2019 [62]	Students were evaluated through publications, presentations, blog posts, and active participation. Regular meetings assessed wellness and professional development.
Navarrete-Welton et al., 2022 [55]	Students were required to write an op-ed for a final project.

**Table 9 ijerph-21-01233-t009:** Studies using self-reported assessment methods following climate-related events training.

Article	Assessment Tools and Evaluation Methods
Basu et al., 2024 [58]	Pre- and post-fellowship surveys used a 7-point Likert scale to assess knowledge, skills, and attitudes. Examples of questions are provided in Appendix 2 of the paper, with responses ranging from 1 (“strongly disagree”) to 7 (“strongly agree”).
Charlesworth et al., 2013 [64]	Awareness and advocacy objectives were assessed using pre- and post-workshop Likert scales. The action objective was assessed through 3-month-post-workshop telephone interviews.
Dunne et al., 2022 [46]	Assessments included pre- and post-module surveys using a 5-point Likert scale to gauge attitudes towards sustainability and self-reported understanding of concepts linked to the learning objectives.
Katzman et al., 2023 [40]	Participant knowledge and self-efficacy were subjectively assessed using a real-time two-question online poll for each session.
Katzman et al., 2022 [42]	Participants underwent subjective assessment through surveys at three points: course registration, post-course, and three months post-course. These surveys measured confidence, knowledge, attitudes, and advocacy using 4-point Likert scales and 5-point agreement scales.

**Table 10 ijerph-21-01233-t010:** Studies recommending assessment methods post-climate-related events training.

Article	Assessment Tools and Evaluation Methods
Boekels et al., 2023 [38]	Various evaluation formats have been established for climate science. Here is the consensus: 1. Examinations should not ask for pure factual knowledge. 2. Traditional formats like multiple-choice exams may not effectively assess transformative competencies. 3. “Deeper” knowledge can be tested through final presentations, reflective writing, special diaries, essays, or complex projects. 4. Project work benefits from allowing students to choose their focus. 5. Competencies can be acquired through smaller transformative practice projects. 6. OSCEs are suitable for climate change teaching.
Evans et al., 2023 [59]	Program assessments varied, with many including post-program knowledge assessments and offering discipline-specific continuing education credits. However, detailed information about the assessment tools was not provided.
Flaten et al., 2023 [60]	One assessment method for measuring students’ readiness to protect public health during disasters and emergencies involved analyzing the impact of extreme weather events on two neighborhoods.
Philipsborn et al., 2020 [56]	The authors recommended various assessment strategies for each content area, such as multiple-choice questions, oral questions, OSCE, clinical evaluations, chart audits, direct observation, standardized patient checklists, and small-group discussions.
Maxwell and Blashki, 2016 [65]	The authors suggested assessing higher-order cognitive skills, affective knowledge, and skills in climate change curricula. Assessments should be specific, relevant, and achievable. Sample assessment items included group presentations, essays, reflective pieces, short answer questions, and placement projects.
Lopez-Medina et al., 2019 [36]	Authors have outlined evaluation methods for sustainability programs, including curriculum and student assessment: 1. Bell (2010) suggests integrated performance assessment with clear outcome identification; 2. Sterling (2009) proposes the 4 Rs model (retain, revise, reject, renew); 3. Koehn and Uitto (2014) advocate the Health Impact Assessment model.

OSCE: Objective structured clinical examination.

**Table 11 ijerph-21-01233-t011:** Studies evaluating student satisfaction with climate-related events courses.

Article	Assessment Tools and Evaluation Methods
Jonas et al., 2023 [41]	Seminar satisfaction was evaluated using a questionnaire, which included yes/no and free-text questions to gather feedback on grading, perceived time and content scope, and the relevance of the three seminar parts.
Charlesworth et al., 2013 [64]	Closed questions explored the importance of climate change to health professionals. Open-ended queries assessed training satisfaction.
Teherani et al., 2023 [57]	Course satisfaction was evaluated through participant surveys, open-ended questionnaires, and post-training interviews.
Katzman et al., 2023 [40]	Program satisfaction was evaluated through a weekly post-session survey on whether the program met the stated objectives, was evidence-based, and increased students’ communication skills.
Jonas et al., 2023 [41]	Seminar satisfaction was evaluated using a questionnaire, which included yes/no and free-text questions to gather feedback on grading, perceived time and content scope, and the relevance of the three seminar parts.
Fülbert et al., 2023 [54]	Elective satisfaction was evaluated through surveys, rating each session from “very poor” (1) to “very good” (5) and providing free-text feedback.

## Data Availability

Available upon request from the corresponding author.

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
