# Peer review of "Core Competencies of the Public Health Workforce in Climate Change and Extreme Weather Events Preparedness, Response, and Recovery: A Scoping Review"

_ijerph, 2024, doi:10.3390/ijerph21091233_

Round 1

Reviewer 1 Report

Comments and Suggestions for Authors

1-      The abstract lacked a structured format, and it did not provide information regarding the eligibility criteria, sources of evidence, or the methods used for charting.

2-      The introduction is composed effectively, presenting a thorough examination of the literature, clearly articulating the objectives, and identifying the existing gaps within the field.

3-      Why is only PubMed used?? Why is Scopus/ISI not used?

4-      In line 185, why ten percent? Please insert reference for this percent.

Author Response

Thank you so much for this review. We appreciate the comments, and have answered them in the attached document. All modifications were simultaneously made within the manuscript.

Reviewer 2 Report

Comments and Suggestions for Authors

This article illuminates a significant gap in the literature and will advance the development of climate curriculum targeted at public health professionals.

The article is vey well-written. It has an organized flow and appropriately uses headers to guide the reader's to key observations. The methodology used for the scoping review is clear. The conclusions are appropriate and provide insights for future areas of study.

There are a few recommendations that I feel might add to clarity and further assist the readers, especially if this is a relatively new field for public health professionals. 

RECOMMENDATIONS

-The abstract should be rewritten. It led me to believe the paper would be about the global impact of climate change for public health professionals who live in maritime settings. Other than the introduction, there is nothing about maritime settings. There is also very little about the global nature of climate change and why public health professionals all over the world need to be prepared. There are examples of changes from Canada, but these focus on ocean impacts. How about the heat domes of Vancouver or the severe wild fire smoke of western Canada? 

-There is very little about the unique learning needs of Public Health professionals. Why do they need something beyond what is currently available in medical or nursing literature? You say that there is a gap, but for non-public health readers, we are left with a sense of "so what?" Why does it matter that curriculum specifically targeted at public health professionals is necessary? I think the introduction needs to be strengthened by citing the depth and breath of climate related issues that impact public health. 

-The concept of planetary health is introduced very late in the paper, yet it is never defined. Planetary health is not a synonym for climate change. It is appropriate in a paper like this, but it needs to be fleshed out a bit. The concept is only 10 years old so for many readers would benefit from the authors providing a definition.

-Search criteria were that papers needed to be published in English or French. Given the global nature of climate change, I think it is important to acknowledge in the Limitations section that there may be papers published in Spanish or an Asian language that could contribute to this scoping review.

Again, this is an important paper and Public Health certainly warrants its own unique set of competencies. I know that because I am a public health nurse, but non-public health professionals may not know that public health is unique from medicine or acute care.

Thank you for the opportunity to see this important work move forward.

Author Response

(The authors gave the same response as above.)
